# Defining a good death: Perspectives of patients, relatives, and health care professionals in the Catalan context—A qualitative study

**Vicky Serra-Sutton** [1,2]*, **Lluïsa Fernández-Giner**[3], **Mireia Espallargues**[1,4], **Anna García-Altès**[1,2,5], **in name of the Good death study in Catalonia**¶

1 Agència de Qualitat i Avaluació Sanitàries de Catalunya (AQuAS), Barcelona, Spain, 2 CIBER en Epidemiología y Salud Pública (CIBERESP), Barcelona, Spain, 3 Independent Consultant, Barcelona, Spain, 4 Xarxa de Recerca en Cronicitat, Atenció Primària i Promoció de la Salut (RICAPPS), Barcelona, Spain, 5 Institut d'Investigació Biomèdica (IIB Sant Pau), Barcelona, Spain

¶ The complete author group of the Good death study in Catalonia can be found in Acknowledgments
* vserra@gencat.cat

**Data Availability Statement:** The narratives of participants and verbatims/quotes supporting the main insights of the study are freely available on

## Abstract

Health systems often lack integrated, person-centered end-of-life care. Improving this care is a key priority for decision-makers in the Catalan context. This study aimed to understand how key stakeholders in the healthcare sector—patients, family members, and healthcare professionals—define the concept of a 'good death' and identify their key common elements in Catalonia, Spain. The research team conducted twenty-three semi-structured, in-depth qualitative interviews with patients receiving palliative care or suffering from complex or advanced chronic conditions, older adults over 65 with chronic illnesses, and their relatives. Three-focus discussion groups were also conducted which involved 31 professionals representing care, management, and planning at a local/regional level as well as representatives from patient and family associations involved in end-of-life care. All discussions and interviews followed a predefined guideline, and we recorded and transcribed them for later thematic content and discourse analysis. Overall, participants did not reach a single, universal definition of a 'good death.' Instead, it appears that a 'good death' involves dying in the way each person desires. In our effort to identify common elements, we found that participants consistently mentioned eight: comfort, placidity, safety, warmth, harmony, intimacy, respect, peacefulness and fulfilment. The study provides valuable insights for service planning and enhances the understanding of the needs of individuals at the end of life, including those with chronic conditions and their relatives, as well as the professionals who care for them and manage services or policies at the local and regional levels in Catalonia. The study highlights participants' preferences for their end-of-life care and their views on death, contributing to a broader understanding of what defines a 'good death.'

the website of AQuAS, the public governmental institution dedicated to evaluating the quality of healthcare and development of decision-making tools/knowledge transfer. For further information, please consult: https://aquas.gencat.cat/web/. content/minisite/aquas/publicacions/2022/estudi_ qualitatiu_bona_mort_annexos_aquas2022.pdf.

**Funding:** The present qualitative study was carried out under a wider project and in the context of the Death Observatory of Catalonia led by the Agency for Healthcare Quality and Assessment of Catalonia (AQuAS) as a commissioned project of the General Directorate of Health of the Catalan Health Department from the Government of Catalonia (Spain) and its general budget (Good Death Study Code: 0002.LE01.20.037). The commissioners had no role in study design, data collection, analysis, or writing of this manuscript. The research protocol and preliminary findings were presented to the Commissioners from the Department of Health of Catalonia to increase usefulness and impact for knowledge transfer and decision-making together with the Steering committee.

**Competing interests:** The authors have declared that no competing interests exist.

## Introduction

In most Western countries, society and culture still consider death a taboo. Health systems often lack high-quality integrated and person-centered strategies in end-of-life care. This deficiency limits the ability to achieve high-quality care standards and to understand the needs of patients and professionals. Additionally, few data are available on the quality of care and people's preferences at this stage of life and around death. In response to these challenges, the Catalan health system set up its Observatory of Death (Motion 91/XI of 2017 of the Parliament of Catalonia urged the government to create this Observatory) in 2019 in Northeastern Spain (see more contextual information in S1 Table). This initiative was created in Catalonia, with the following objectives: a) record and provide data on end-of-life care; b) raise public awareness of preferences regarding where, with whom, how, and when individuals wish to die; c) compile information to draw conclusions and propose improvements in the end-of-life process; d) help break the existing taboos surrounding death [1,2].

The need for measures and practices to improve the quality of end-of-life care is a priority for decision-makers within the Catalan context. In this sense, the Observatory's Steering Group also identified significant gaps in managing end-of-life and death. As a result, the Strategic Chronic Health Plan and Intermediate-Long Care Strategic Health Program in Catalonia in 2021 agreed to conduct a qualitative study to explore what is considered a good death. Describing the elements of a good death would be a step forward to continue breaking taboos around death in order to define improvement actions in health care and planning.

Several scoping reviews have proposed definitions of the term "good death" and identified key domains prioritized by patients, relatives, and healthcare professionals who take care of the patients [3–7]. However, consensus on what a "good death" is or on the needs of patients in the final stages of life, remains elusive. For instance, the definition published by the Institute of Medicine in the United States in 1997 [8] has faced criticism for lacking external standards and being overly reliant on individual perspectives. It describes a good death as one free from avoidable anxiety and suffering, aligned with patients' and families' wishes and, consistent with clinical, cultural, and ethical standards.

Even though publications have recorded the perspectives and opinions of different stakeholders, there is still a relative lack of information on the experiences of people who are not hospitalized and still live in their homes. Often, the caregivers are the ones who act as interlocutors to assess the users' experiences of the services and the care received. Some studies have argued that the opinions of relatives are often recorded once the person has died, not during the process of dying or at the end-of-life phase of the patient. In short, the consulted documents note that the views of the people who die or wish to die often go overlooked and identify barriers in health systems at end-of-life.

There is remaining lack of data on preferences for where, how, and when to die. So, a qualitative study was designed to deepen in the complex concept of a "good death" through the experiences and views of individuals with chronic health conditions, relatives, and healthcare professionals in Catalonia. Initial discussions with the Steering Group highlighted the understanding that the meaning of a good death may differ between younger and older adults and can change over time, particularly as one approaches the end of life. Finally, the research team realized that, to reach an ideal "good death," people needed to have a plan in advance, and both relatives and professionals had to respect their preferences and anticipated decisions. Thus, this project's results would aid in understanding and sharing knowledge of the needs, best practices, and possible improvements needed in the Catalan health system. These insights could inform decision-makers across clinical, care, management, and policy levels in both

governmental and non-governmental organizations, to define strategies/actions to improve end-of-life care and understand people's needs around death.

In global terms, the qualitative study was conducted to understand in depth the perspectives, opinions, and experiences of the above-mentioned groups, focusing not only on the concept of a good death and its meaning and also on barriers, facilitators, and unmet needs within the system to get closer to the ideal "standards" or "preferences" for a good death. Given the complexity and subjective nature of the topic, a mix of techniques, theoretical-methodological approaches, informants, and research perspectives were employed to reach a deeper understanding of the meaning given to a "good death" in Catalonia.

Two primary techniques were applied: 1) Semi-structured in-depth interviews; 2) focus group discussions, which included a selection of images representing a good death for participants. Regarding participants, three main stakeholder types were involved: patients, relatives, and professionals. Additionally, the following theoretical approaches were applied: 1) Phenomenological and hermeneutical perspectives: to capture the individual meanings and experiences related to a good death; 2) a socio-constructivist perspective to explore how individuals construct their reality taking into consideration that change in knowledge or meanings can be expected through added information and empowerment [9–12].

We considered common characteristics based on daily living, experiences, and feelings of participants (patients and relatives) and the subjective definition of the meaning of a good death, which comes from a learning and socialization process (phenomenological approach) [9,12] Furthermore, we wanted to understand the individual experience of participants, and the events and relationships considering their social, cultural, and historical (life-course) context, going beyond a description of what they manifest, to reveal hidden meanings and interpret them (hermeneutical approach) [12,13]. Finally, we studied the meaning of what a good death is through the collective generation of meanings, using language and interaction to produce social meanings of this construct/phenomenon. We assumed that people construct their reality and can change it with more information and empowerment (socio-constructivist approach) [9,12].

The scope of this paper includes part of a wider qualitative study. In the present paper, we propose to answer the following research questions: 1) what is the meaning of a good death for a selection of people living in Catalonia with a chronic health condition, a close relative, and for multidisciplinary health professionals? 2) How do patients, relatives, and professionals understand what a good death is in the Catalan context in general terms? 3) What common elements should a good death include for different types of participant profiles? 4) What differences do diverse participant profiles show in their opinions about what constitutes a good death?

Ultimately, this study aims to deepen the understanding of the perceptions of the term "good death" as expressed by patients, relatives, and professionals within the Catalan health system. Establishing the attitudes of these key stakeholders toward this complex and subjective concept is an essential first step toward proposing solutions for improving the final stage of life.

## Materials and methods

This qualitative inductive study used semi-structured in-depth interviews with patients, relatives, and focus group discussions with healthcare professionals involved in care, management, planning, or patient associations. The study, as mentioned before, applied three theoretical-methodological approaches: phenomenological, hermeneutical and socio-constructivist perspectives, in two phases [9–12]; more details can be found in S1 Table] We triangulated data

from different sources of participants, techniques, and approaches to capture key stakeholders' experiences and opinions (S2 Table includes the topics in the guideline for the whole project).

## Phase 1: Semi-structured interviews

In Phase 1, we conducted semi-structured in-depth interviews with patients and relatives in their homes or at healthcare centers for institutionalized participants (intermediate or long-term care facilities). This approach aimed to capture their daily living context and ensure their comfort [12]. Each interview lasted between 90 and 120 minutes and was conducted by a senior qualitative researcher using a predefined guideline agreed upon by the research team and steering committee.

The first part of the interview included a selection of pictures of nature used as an ice-breaker to help explain the representation and meaning of a good death. After obtaining written informed consent, all the interviews were audiotaped for transcription and subsequent analysis. Key insights from each interview were discussed by two researchers, and anonymized summaries were shared with the Steering Group to ensure confidentiality.

## Sampling

We selected a theoretical, planned, reasoned, and stratified sample [12,13] of 25 patients and relatives and divided them into four segments (based on health status and life stage; see more details in S1 Table). This sample included people with a variety of health diagnoses and comorbidities, and sociodemographic characteristics (age, sex and residence-urban/rural, health region, and socioeconomic status). The sampling was stratified and purposive, in an attempt to obtain a wide range of discourses. The Steering Group of the study, the Consultative Council of Patients of Catalonia, and the Catalan Health Institute collaborated in the recruitment of patients and relatives. The Agency for Health Quality and Assessment of Catalonia (AQuAS) coordinated the recruitment and fieldwork with the support of the following organizations and institutions:

- Association against Cancer (AECC)

- The Catalan Association for the Right to Die with Dignity

- Miquel Valls Foundation, Catalan Foundation for Amyotrophic Lateral Sclerosis

- Catalan Association of Patients with Advanced Respiratory Disease and Lung Transplantation (AIRE)

- Catalan Association for Parkinson's Disease

- The Kreamics Association of Burn Victims of Catalonia, Family and Friends

- Management and Provision of Health Services (GIPSS) in Tarragona

- Primary Care Team from the Catalan Institute of Health in Camp de Tarragona

   The general inclusion criteria were the following:

- Adults (18+) living in Catalonia, who presented complexity associated with a limited life expectancy (classified under the acronym in Catalan of MACA: advanced chronic disorder), [9] complex care needs, or complex chronic conditions (classified under the acronym PCC: complex chronic patient) [14], or

- Adults (65+) living in Catalonia, but not classified as MACA or PCC with a possible chronic condition, or

- Relatives of people classified as MACA or PCC over 18 years of age, living in Catalonia.

     The general exclusion criteria were the following:

- People unable to participate due to auditory or cognitive limitations or illness.

- People uncomfortable discussing the topic at the interview.

- People who, at the time of the interview, were working in the field of public or private health-care or end-of-life care.

     People who agreed to participate and share their opinions and experiences, signing a consent form, were contacted to arrange an interview at their home (or institution where they were admitted; e.g., intermediate and long-term care).

## Phase 2: Focus group discussions

The research team held three virtual focus sessions with 10–12 participants via a corporative Zoom of AQuAS. The sessions involved health and social care professionals, and professional representatives from of patient and/or family associations [13,15]. Each session lasted between two and a half and three hours, applying pre-defined guidelines agreed upon by the research team. The first part of the focus group discussions included choosing a photo/image of nature to describe what a good death was for each participant. Then, the main insights and themes from Phase 1 were presented and discussed to obtain further elements and understanding of the meaning of what a good death is. A senior researcher along with the coordinating researcher moderated these focus groups. All focus discussion groups were taped after obtaining the participants' consent and written authorization. The study team transcribed and analyzed the sessions, ensuring that any identifying information from the participants was excluded.

## Sampling

The study´s Steering Group aided in the recruitment coordinated by AQuAS of a theoretical, planned, reasoned, and stratified [12,13] sample of 35 professionals related to end-of-life care from primary and home care, hospitals, intermediate care, nursing homes, and patient associations. The recruitment considered findings from a literature review and stratified the sample into four segments based on professional backgrounds (see S1 Table).
     The general inclusion criteria were the following:

- Professionals providing direct care to people in the final stages of life, in palliative care, including specialists in geriatrics, neurology, cardiology, oncology, pneumology, surgery, psychiatry, internal medicine or family and community medicine, nursing, psychology, and social work.

- Professionals involved in clinical management or health planning for end-of-life care. Researchers focused on end-of-life and death studies.

- Representatives from patient associations dealing with complex chronic conditions.

     The exclusion criteria were the following:

- Age under 18; inability to take part actively in the discussions and generation of ideas due to sensory or cognitive limitations.

- Participants lacking sufficient proficiency in Catalan or Spanish for discussion.

- Unease in discussing the study topic.

Participants needed a reliable internet connection to engage in discussions lasting 2½ to 3 hours.

## Fieldwork and topics recorded

The research team recruited participants from December 2021 to March 2022 and the field-work took place from January to April 2022. This included interviews with patients and relatives and focus discussion groups with professionals. S2 Table outlines the topics covered during the interviews, adapted to each informant type (patient or relative). Additionally, the team shared the main insights from the semi-structured interviews with professionals in Phase 2. They recorded participants´ residence (defined as urban or rural), as well as the health region in Catalonia in which they lived or worked.

## Data analysis

The researchers carried out an inductive thematic content and discourse analysis (see more detail of the rationale in S1 Table (points 17 and 18) [9,12,15,16]. The thematic content and discourse analysis was carried out for sample segments and globally for each relevant topic [15]. The transcriptions were read and re-read. During analysis participants' social backgrounds were considered to understand their perceptions of a "good death", taking into account social constructions of its meanings. The researchers explored commonalities and differences among participants' narratives and themes (elements of a good death). The themes and insights from semi-structured in-depth interviews (Phase 1) were triangulated with those that emerged from the narratives in the focus group discussions (Phase 2). Then, they were discussed by the core research team and subsequently with the Steering Group to confirm their coherence and robustness. Vivid quotations were selected to reinforce the themes and insights shown, adding brief information of the profile of participant to understand a little of their context.

## Ethical and legal considerations

The research team adhered to the AQuAS code of good scientific practices during all the qualitative study. They obtained the necessary consents and authorizations required internally by AQuAS and the research protocol was approved by the Research Ethics Committee of the Jordi Gol i Gurina University Institute for Research in Primary Health Care Foundation (IDIAPJGol, 21/251-P, 24 November 2022). The study followed the regulations and the data protection and security policies of the Catalan government and AQuAS. Moreover, the data provided by the participants in the qualitative study were treated under the legislation on data protection in force, including EU Regulation 2016/679 of the European Parliament and Council of 27 April 2016 on data protection. All data were stored to reinforce anonymity and privacy following the Spanish Law on Data Protection (LOPD).

## Results

### Characteristics of participants and general understanding of a "good death"

This qualitative study included a total of 54 participants consisting of patients, relatives, and professionals ("Table 1"). The final sample did not include professionals focused exclusively on research. Most participants who were involved in the focus group discussions worked in direct care or clinical management, and some also involved in research.

**Table 1. Characteristics of participants in the qualitative study.**

| Participants' profiles | n (%) |
|---|---|
| Patients and relatives (n = 23) | |
| *Type of profile & health status* | |
| *Patients with advanced chronic disorders* | 6 (26.1) |
| *Patients with complex chronic care disorders* | 8 (34.8) |
| *Patients over 65 without advanced or complex chronic disorders* | 4 (17.4) |
| *Relatives* | 5 (21.7) |
| *Gender* | |
| *Female* | 14 (60.9) |
| *Male* | 9 (39.1) |
| *Age* | |
| *26–45 years old* | 3 (13.0) |
| *46–64 years old* | 3 (13.0) |
| *65–79 years old* | 16 (69.6) |
| *>80 years old* | 1 (4.3) |
| *Place of residence* | |
| *Urban* | 19 (82.6) |
| *Rural* | 4 (17.4) |
| | |
| Professionals (n = 31) | |
| **Profile** | |
| *Health care* | 18 (58.1) |
| *Health management & planning* | 8 (25.8) |
| *Patients associations* | 5 (16.1) |
| **Gender** | |
| *Female* | 25 (80.6) |
| *Male* | 6 (19.4) |
| **Health sector** | |
| *Palliative care program* | 8 (25.8) |
| *Acute specialized hospital* | 8 (25.8) |
| *Primary health care* | 5 (16.1) |
| *Patient & family associations* | 5 (16.1) |
| *Residential care* | 3 (9.8) |
| *Emergency services* | 1 (3.2) |
| *Governmental institution* | 1 (3.2) |
| **Region represented** | |
| *Barcelona* | 11 (35.5) |
| *Girona* | 7 (22.6) |
| *Lleida* | 3 (9.7) |
| *Tarragona* | 6 (19.3) |
| *Whole of Catalonia* | 4 (12.9) |

Participants reported that a prerequisite for a good death—whether for themselves or their relatives—was having lived a good and long life, while maintaining an acceptable quality of life (more details of selected quoting can be found in S3 Table). For this reason, one could argue that experiencing a minimum level of quality of life before death is essential for achieving a "good death." As one participant stated: *"It's great that you're carrying out this study to find out what a good death means to people, but it's pointless if we don't first establish what it means for us to have a good life"* (Patient with a complex chronic condition).

The reflections on "quality of life" imply having a level of autonomy and the ability to enjoy experiences during the time before death. It should also be stressed that both autonomy and enjoyment are subjective perceptions and, as such, can change and adapt to new circumstances. That is to say, a clinical condition that severely limits autonomy or enjoyment of life, -e.g., losing the ability to speak again due to the need for a tracheotomy required in the palliative treatment of amyotrophic lateral sclerosis- might not be as restrictive -if, for example, the patient can communicate through other means, such as a computer (see quotations in S3 Table).

### Common elements of a "good death"

While nearly every interviewee offered a unique definition of a good death, one shared idea emerged: a good death consists of *"dying in the way each person wishes"* (patient, advanced chronic condition). Despite this individual subjectivity, eight common elements were identified (see *Fig 1 and S3 Table*), which reflect and summarize what might be considered a "good death." These elements can be combined to form a cluster of what a good death means for each respondent (their own, or that of a relative/close friend), in terms of how, where, and with whom they want to die.

Of these eight elements, the most universal and frequently mentioned was **comfort and placidity**, understood as the complete absence of physical suffering, such as suffocation and pain, and mental suffering, such as anxiety and distress. This state also involves a body-mind sense of well-being similar to that experienced in the transition from being awake to falling asleep, in which, in a matter of seconds or minutes, the mind gradually disconnects from its external stimuli and enters a state of "peacefulness".

| | | | |
|---|---|---|---|
| -Absence of physical (suffocation, pain) and mental (distress, anxiety) suffering.<br>-Feeling of absolute and comprehensive wellbeing similar to the state experienced in the transition from being awake to falling sleep. | -Assurance that there will be no complications in the person's physical and mental state that could cause unnecessary suffering; also assurance of receiving continuous medical/ clinical care that is needed. | -Feeling of love or affection and emotional containment when accompanied by significant others.<br>-Feeling well treated by attending professionals. | -Perception of good group synergy; no disruptions or conflicts among the people accompanying them. |
| **Comfort and placidity** | **Safety** | **Warmth** | **Harmony** |
| -Being in a private and quiet space without disruptive external stimuli or third parties. | -The perception that the professionals and relatives respect their wishes (or those of the designated representative). | -Feeling of internal harmony, which includes the acceptance of one's death and the absence of any unresolved conflicts ("being at peace with oneself and with others"). | -A feeling of satisfaction with the life lived, which includes believing that it has been worth living, and that a minimum quality of life is maintained until death. |
| **Intimacy** | **Respect** | **Peacefulness** | **Fulfilment** |

**Fig 1. Eight elements that summarize what constitutes a "good death" in the qualitative study with patients, relatives, and healthcare professionals in Catalonia (Spain).**

*"A good death is when "I fall asleep and I don't notice [that I'm dying]". . . it´s like the moment when "I go to sleep and I don't wake up," very closely related to what you mentioned as the moment you fall asleep [. . .]"* (Professional, social worker).

*"I don't think it's worth living if you're suffering," "It´s not that I prefer death, but I prefer not to suffer," "It´s different to die from something like a heart attack than to die after a long illness," "if I need to be intubated (for life) and be like a vegetable without noticing anything, I'd rather have an injection. . . And goodbye!"* (Patient, complex chronic condition, between 65 and 79 years).

The second element, **safety**, ties into the first; since it reflects the need to feel reassured that comfort and placidity will be sustained throughout the dying process. This reassurance comes from medical support, ensuring that drugs like analgesics and tranquilizers are readily available when needed.

The third element highlighted by most patients and relatives was **warmth**, the feeling of being accompanied emotionally by someone with whom the dying person has a significant bond and who helps them in the transition from life to death with affection and care. Kind treatment from healthcare professionals also played a key role in creating this sense of warmth.

The fourth element, **harmony,** relates to warmth and involves the ability of the people accompanying the person to react with honesty and calmness, **"*and not make a fuss.*"** It also requires the ability to maintain a positive and conflict-free relationship. Quoting one patient, *"They mustn't fight over whether or not they should keep me alive, or about the treatment I should have."*

The fifth element is **intimacy**: for a good death, the person needs to have a private space, whether they are alone or with others [see S3 and S4 Tables]. Care must be taken to ensure that there are no distressing external stimuli (for example, screams from the next room) and no third parties or outsiders who might spoil this intimacy (for example, the person dying should not be in the same room as another patient).

The sixth element is **respect** for the wishes of the dying person (or the designated representative). This is a kind of "fundamental right," especially in cases in which an advance directive (colloquially known as a "living will") has been made, or if the person has verbally expressed to their relatives their decisions regarding certain treatments or medical practices or the circumstances in which they might opt for euthanasia. This respect is considered essential by accompanying relatives, but above all, by the physicians and nurses attending the person at the time of death.

The seventh element, mentioned by patients and relatives, is peacefulness. Peacefulness involves the sensation of being at peace with oneself and with others, the feeling that no important matters or conflicts have been left unresolved, and the acceptance of one' own death or that of a loved one. Some patients and relatives note the key role of spirituality or religion in achieving this **peacefulness.**

Finally, the eighth element is fulfillment or, the feeling of having lived a life full of significant emotions and experiences. It involves achieving a quality of life that allows a person to depart this world with satisfaction and to leave behind good memories for loved ones.

## Relevant elements of a "good death" for relatives

Relatives view a "good death" for their loved ones as one that is positive and free from trauma, flowing smoothly. Therefore, their focus tends to be on the variables of comfort and placidity, safety, warmth, and intimacy (additional quotations and narratives can be found in S3 Table). For relatives, the most important point is to ensure that their loved ones do not suffer in any

way, feel accompanied and valued, and can pass away in a quiet and private space (S1 File). In any case, as noted above, each person (in this case, each relative) has their particular cluster of elements that comprise their concept of a "good death" (in this case, thinking of their sick relative), and they often include other elements (Fig 1)–above all, harmony and respect, and, less frequently, peacefulness and fulfillment.

## Key elements of a "good death" for professionals

Healthcare professionals perceive a "good death" similarly to patients and relatives, considering both the moment of death and the life leading up to it. In the focus group discussions, professionals mentioned several aspects that define a good death. They consider it important to ensure comfort and avoid suffering—understood as providing the person with all the available tools and necessary therapeutic means. Healthcare professionals highlighted the importance of receiving proper training to ensure that the person does not suffer and feels protected and safe throughout the dying process, while also allowing for an intimate experience. However, unlike patients and relatives, professionals did not tend to evoke the idea of "placidity" or agree that a good death implies feeling in a state like that of the transition between being awake and falling asleep.

> "I *think peacefulness and internal harmony are key points. It´s about being at peace with yourself. However, some patients face a disease with a rapid decline and. . . find it hard to accept the end of life, making it challenging to achieve this peacefulness. A necessary first step is acknowledging,* "*I've done everything I can; I've reached this point." While primary care services and hospital services obviously do all they can, attaining peacefulness, but acceptance is a necessary step"* (Professional, primary care physician).

> "*People who die with someone next to them, I mean, with a patient next to them. . . I mean, this happens. . . I guess it doesn't happen in acute care hospitals in general anymore, because awareness is growing and you [other professionals participating in the focus group discussions] are working hard on this issue, but there are still areas in which it happens. In addition, this, in the order of priorities, is the most serious thing that can happen; I mean sharing a room with another person who is not at the end of their life"* (Professional, patient association).

Another priority that emerged from the discussion with professionals to achieve a good death was the need for emotional support, understood as the provision of warmth and company by both practitioners and relatives. This was particularly important in light of the experience of the COVID-19 pandemic in which thousands of patients died alone or were accompanied only by healthcare personnel. In this sense, no matter how much effort is made by the health staff to provide accompaniment and support for patients–including, on occasions, connecting them with their relatives through a video call–, the pain and suffering generated by the restrictions on visits compounded the impact of the disease itself. In any case, the professionals stressed how important it was that the loved ones could accompany patients throughout their illness and, particularly, at the moment of death. The results also draw attention to the key role of healthcare professionals from different disciplines in providing empathy, warmth, and emotional support.

> "*I come from an association, from a patient organization. What are we finding? That many people approach us, especially relatives, looking for something that they have not found in the healthcare services. Many times. . . I come to see if you [referring to other care professionals in the focus group discussions] can help me, because I don't know who to talk to, and I'm*

*looking for the psychosocial and emotional support that these relatives miss–and that's why they come to us for assistance*" (Professional, patient association).

"*I don't know if the home care program and support team* [Spanish and Catalan acronym PADES] *are enough;* [. . .] *if they are sufficient, I don't know, but it should be available to everyone, I think that this is very important. . . They should invest in these teams; let them work with dignity and with enough resources.* [. . .] *I would say that having a good death is having a good accompaniment*" (Professional, patient association).

"*Being able to have spaces for communication, having space or being accompanied, being able to speak in case of need. In the case of family members, patients are very often more aware of their situation than their relatives, and they lack someone to share their fears, doubts or unresolved issues. So, for me, a good death would also be to have somewhere, and someone (ideally a family member, but if not, a professional) whom I can talk to openly and feel that I'm being listened to, that I'm not being judged and that there is this safe space*" (Professional, social worker).

Sadly, professionals sometimes witness patients dying in the hospital corridor, in a cubicle within emergency units, or sharing a room with another patient. However, there are also testimonies of professionals who have seen patients dying in privacy despite being in a busy hospital environment. The importance of feeling peacefulness and fulfillment (as understood by patients and family members) is also echoed by professionals, who emphasize the acceptance of death and the importance of being able to leave fond memories among those who remain.

## To plan or not to plan: a key point to understanding the implications of a "good death"

A sudden death–understood as a death that occurs unexpectedly and lasts only a matter of minutes–tends to be considered the paradigm of a good death, or the "ideal" death. This is because the person "does not suffer so much physically or psychologically" (e.g., due to a fulminant heart attack) and they die in a state of calmness–typically, as dying in their sleep. However, a deeper examination reveals a significant tension between the desire to die suddenly, without knowing it, and the desire to be able to anticipate one's death to "prepare" for it and "leave everything in order". "Preparing for death" involves activating mechanisms that help people connect with others and with their inner spiritual worlds, and to die in peace with others and oneself–goals over which control is lost if death comes suddenly:

"*A good death is when 'I fall asleep and I don't notice'* [that I'm dying]*. . . as the moment when 'I go to sleep and I don't wake up,' very much related to what you mentioned as 'when you are falling asleep'.* [. . .] *Taking into account the taboo of death for society, how we approach the whole issue of suffering from childhood to adulthood, sometimes dying suddenly means not having to say goodbye to family members, not having to solve pending issues, not having to face that moment.* [. . .] *Dying suddenly avoids the whole process associated with other types of death*" (Professional, social worker).

"*I think a good death would be a heart attack that results in immediate death. That is the best, I think, from my point of view. . . Because you are not sick, there is nothing wrong and, suddenly, on the same day it happens to you. . ., and that is it* [. . .]" (Relative, woman, 46 to 64 years old).

"*I want to die on a comfortable mattress, sleeping peacefully. The only thing that frightens me a little is not being able to die when I want*" (Patient, woman, 65 to 79 years old).

As seen in some of the above quotes, some participants consider a "good death" to be the moment of a placid transition similar to falling asleep:

"**I** *just wanted my mother not to suffer, and she didn't. She fell asleep and that was it; she died like a candle going out. . . and that is it. The most beautiful death a person can have. The death of my mother*" (Patient, over 65 years old without chronic health conditions).

## Discussion

This study has deepened our understanding of what a good death is for patients, relatives, and professionals in Catalonia, and has identified a set of common elements that mainly reflect the need to honor people´s wishes and preferences at the end of life. This can be extrapolated to other regions in Spain and Europe. In Catalonia, a series of specific tools have been developed to guide the design of public health and clinical management, as well as research to acknowledge the preferences of people around end-of-life care. Examples include NECPAL, a tool for improving care and managing patients' preferences, which evaluates the need for palliative care and provides a prognosis of end-of-life [16], or the advanced healthcare directive and the advanced care plan [17]. Despite these advancements, there is still a lack of robust quantitative data on the methods and preferences related to dying [S1 File, 1,2], as well as insufficient awareness of patients' needs. Consequently, this study is particularly timely, emphasizing the critical need to address these gaps to improve end-of-life care.

In general, our study corroborates the results of published literature in different international contexts: there is no single universal definition of a "good death" [1–5]. To provide an operational definition that might lead to improvements in the healthcare system, we can identify a series of common elements that capture this subjective and complex phenomenon, more precisely than defining specific standards and actions to improve service quality. In a systematic review by Krikorian et al., the core elements of a "good death" included control of pain and symptoms, clear decision-making, feeling a sense of closure, being seen and perceived as a person, preparation for death, and being still able to give something to others [4; S4 Table]. In specific populations such as adult patients with cancer in a palliative phase [18], a "good death" was associated with living conscientiously in the face of imminent death, preparing for that death, and dying in comfort (e.g., dying quickly, with independence, with minimal suffering and with one's social relations intact). Takahashi's scoping review showed that the characteristics of a good death in dementia were similar to other populations and identified specific features, such as the need to be surrounded by familiar things and people, and person-centered communication at the end of life. [6] Zaman et al.'s systematic review detected similar elements and key themes to the ones that we found in the present study, such as dying in a preferred place, relief from pain and psychological distress, emotional support from loved ones, autonomous decision-making regarding treatment, avoidance of futile life-prolonging interventions and being a burden to others, the right to assisted suicide or euthanasia, and effective communication with professionals [7].

Promoting a "good death" is a key component of end-of-life care and palliative care strategies [14,19–21]. However, many aspects of end-of-life care need improvement including the ability to identify when a person is nearing death, the interventions aimed at alleviating physical and psychological pain, and the support offered to families in coping with and accepting the situation. Healthcare professionals also require training in the communication skills necessary to provide quality support to patients and families without fear or taboos.

The guidelines and action protocols of reference institutions such as the National Institute for Clinical Excellence (NICE) and the Institute of Integrated Health and Social Care in the

United Kingdom have described criteria based on clinical strategies and health planning to facilitate a good death and thus provide a better quality of end-of-life care. For example, the National End of Life Care Strategy for England, published in 2008 [20], highlights the following key elements of a good death: being treated as an individual, with dignity and respect, being without pain and other symptoms, in familiar surroundings, and accompanied by close family and/or friends. In this sense, a report from the Organization for Economic Co-operation and Development (OECD) [22], recommends that more information on end-of-life care should be provided for healthcare staff, along with improved evaluation processes, to ensure that people's wishes are respected and that quality of care can improve.

In a society where there is widespread reluctance to discuss death openly, the professionals who participated in the present qualitative study emphasized that, even when a patient is not suffering and the proposed eight elements of a good death are present, "we still want to die quickly, because we don't know how to die." Death is no longer integrated into everyday life, as it was in our grandparents' generation; we live our lives at a much faster pace and we are far less patient than our predecessors. Professionals also criticize the lack of training for handling death; they report that the focus at medical schools is on "teaching how to heal and save lives," applying treatments that prolong life; the death of a patient is regarded as a failure. In line with the OECD report on the needs of people at the end of life [22], Kawaid et al.'s meta-synthesis [23] highlighted the importance of providing healthcare staff with more information about end-of-life care and improving evaluation methods to ensure that patients´ wishes are respected and that the quality of care is enhanced.

The main limitation of this study is the potential for selection bias. The interviewees were all people who were ready to talk about death and the right to die with dignity; they do not consider the experience of death to be taboo. Additionally, there were some imbalances in the recruitment of the patient and relative samples, and we cannot determine the extent to which these imbalances may have distorted the study's main insights. Accessing subgroups of people living alone, middle-aged or younger people, and additional people with lower socioeconomic resources would have provided more information regarding attitudes towards how, where, and with whom people prefer to die. One possibility now would be to replicate this study in a radically different target population, for example, by including children and adolescents among the sample of relatives or other profiles of people affected by death processes. However, despite the mentioned limitations, we think that with the current sample in this two-phase study, we have obtained sufficient information with saturation of the elements of a good death found. Probably, the needs, barriers, and facilitators will vary across different contexts and this should be studied in the future.

## Conclusions

The study confirms that achieving a single, universal definition of a "good death" is challenging, as people's perceptions are subjective and unique, much like their definitions of a "good life." Despite this subjectivity, eight common elements emerged from the interviews and focus discussion groups. Individuals can prioritize one or more of these elements, creating a personalized vision of their ideal "good death." The eight elements identified are comfort and placidity, safety, warmth, harmony, intimacy, respect, peacefulness, and fulfillment.

All participants, including professionals, agree that the absence of pain and distress is essential for achieving a good death. The wish to die "without suffering" is expressed across all groups and aligns with the findings from other studies on the meaning of a good death. However, this "dying without suffering" refers not only to the moment of death, but also to the importance of having lived a fulfilling life, with an acceptable quality of life and possibly

enduring "bearable" suffering until the moment of death. Once again, the distinction between "bearable" and "unbearable" suffering is subjective and unique to each individual and is a perception that can change over time.

## Supporting information

**S1 Table. Conceptual, theoretical, and methodological considerations of the qualitative study on good death in Catalonia.**
(PDF)

**S2 Table. Topics raised in in the in-depth interviews and focus group discussion sessions.**
(PDF)

**S3 Table. Quotes from participants that reinforce the eight core elements for a good death.**
(PDF)

**S4 Table. Core elements of a good death, participants, design of this study and other selected studies on the topic.**
(PDF)

**S1 File. Additional insights on the preferences for the place to die as a facilitator for achieving a "good death".**
(PDF)

## Acknowledgments

**The complete author group is the following: Serra-Sutton V, Fernández-Giner LL, Espallargues M, García-Altès A, in name of the Good death study in Catalonia (Barbero-Biedma E, Busquets-Font JM, Cantarell G, Carreras-Marcos B, Costas-Muñoz E, Delagneau-González J, Lasmarías-Martínez C, AA. Qanneta R, Sánchez-Martorell A, Santaeugènia- González S, Selles E). The present study is part of a knowledge transfer project commissioned by the General Health Directorate of Planning and Research from the Department of Health in Catalonia developed in the context of the Observatory of Death coordinated by AQuAS. The Steering Group of the Good Death Study in Catalonia contributed to the discussion of study guidelines, sampling design, main insights and also the dissemination of findings. The authors thank all participants for sharing their views and experiences. We would also like to thank Rosaura Vidal of the Association against Cancer of Catalonia and the Consultative Council of Patients of Catalonia for her support in the process of recruiting participants for the study. The authors are grateful to Antoni Parada from AQuAS for his support in the ad hoc literature searches and communication strategies, to Michael Maudsley for the translation into English of a previous version of this paper, and to Mónica Caldeiro and Sabrina Silano from Vitruvian Medical Communications for their final review.

## Author Contributions

**Conceptualization:** Vicky Serra-Sutton, Anna García-Altès.

**Data curation:** Vicky Serra-Sutton.

**Formal analysis:** Lluïsa Fernández-Giner.

**Funding acquisition:** Vicky Serra-Sutton, Mireia Espallargues, Anna García-Altès.

**Investigation:** Vicky Serra-Sutton, Lluïsa Fernández-Giner, Anna García-Altès.

**Methodology:** Vicky Serra-Sutton, Lluïsa Fernández-Giner.

**Project administration:** Vicky Serra-Sutton.

**Resources:** Vicky Serra-Sutton, Mireia Espallargues.

**Supervision:** Mireia Espallargues.

**Validation:** Vicky Serra-Sutton, Mireia Espallargues.

**Visualization:** Vicky Serra-Sutton.

**Writing – original draft:** Vicky Serra-Sutton, Lluïsa Fernández-Giner.

**Writing – review & editing:** Mireia Espallargues, Anna García-Altès.

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
