## [Decision Letter · Decision Letter 0]

2 Aug 2023

PONE-D-23-14275What is a good death for patients, families and health professionals? A qualitative ethnographic and phenomenological study in the Catalan contextPLOS ONE

Dear Dr. Serra-Sutton,

Thank you for submitting your manuscript to PLOS ONE. After careful consideration, we feel that it has merit but does not fully meet PLOS ONE’s publication criteria as it currently stands. Therefore, we invite you to submit a revised version of the manuscript that addresses the points raised during the review process.

We look forward to receiving your revised manuscript.

Kind regards,

Martin Mbonye

Academic Editor

PLOS ONE

Additional Editor Comments:

Dear Dr. Serra-Sutton,

Thank you for submitting your manuscript to PLOS ONE. After careful consideration, we feel that it has merit but does not fully meet PLOS ONE’s publication criteria in its current form. Therefore, we invite you to submit a revised version of the manuscript that addresses the points raised during the review process.

The manuscript has been evaluated by two reviewers, and their comments are available below. The reviewers have raised a number of concerns that need attention. A common comment from both reviewers is on the disconnect between the introduction and the methodology which you have to address. Pay close attention to the comments on the methods section in general pointed out by both reviewers. Reviewer 2 was particular about the language editing that you need to pay attention to. Could you please revise the manuscript to carefully address the concerns raised?

We look forward to receiving your revised manuscript.

Kind regards,

Martin Mbonye

Academic Editor

PLOS ONE

Note: HTML markup is below. Please do not edit.]

Journal Requirements:

"The study is part of a governmental commissioned project of the General Directorate of Health of the Catalan Health Department from the Government of Catalonia (Spain) in the context of the implementation of the Death Observatory to the Agency for Quality in Healthcare and Assessment of Catalonia (AQuAS).  "

Reviewers' comments:

Reviewer's Responses to Questions

Comments to the Author

1. Is the manuscript technically sound, and do the data support the conclusions?

Reviewer #1: Partly

Reviewer #2: Partly

2. Has the statistical analysis been performed appropriately and rigorously?

Reviewer #1: I Don't Know

Reviewer #2: N/A

3. Have the authors made all data underlying the findings in their manuscript fully available?

Reviewer #1: No

Reviewer #2: Yes

4. Is the manuscript presented in an intelligible fashion and written in standard English?

Reviewer #1: Yes

Reviewer #2: Yes

5. Review Comments to the Author

Reviewer #1:

When looking at the text, I don’t see any obvious red flags. I, however, don’t see the information in the text that I require to be able to make an adequately detailed decision. I was hoping that I would find, attached somewhere, an annex that would allow me to understand exactly what they had done and why. I do believe that at the end of my review process I will come to the conclusion that what I’m seeing appears to be sound, but that the initial structure of the article does not adequately anticipate the methodology section, so it requires substantial revision. What I mean by this is that the choices made in the methodology section require information which should be, but is not present in the introduction. I also believe that I will come to the conclusion that the method which they used is not a discourse analysis. As I have often found, I suspect that what I will find, when I have the opportunity to review more detailed information, is that they did a sound thematic contact analysis. It is a constant frustration of mine that authors in my field, feel, properly, compelled to claim that they are doing analysis methods which are far more complex than that either possible or necessary for their purposes.

I will find that the article failed to follow the expectations of annotation for transparent inquiry, which means that I do not understand the logic which they have used in sampling quotations for presentation in their text. I will also find that they failed to remain consistent in their recognition that the object, good death, would be understood very differently by their respondents. What this means, in consequence, is that rather than present, a set of scenarios at the end, each some sort of idealized representation selected for centrality. What they have done is present a list of common elements across respondents that are chosen more for commonality than salience. This sort of presentation of an intersection of subjective constructions is a common strategy that is used which suffers the unfortunate characteristic of failing to remember that each of the construction is considered may fail horribly when shorn of attributes specific to that single construction.

At the end, what I think I will find is that the article does an interesting examination of stakeholders understandings of a very complex and subjectively constructed construct, the good death.

Reviewer #2: The manuscript would benefit from language editing. In some cases, the language needs to be checked to suit the standard language used in research.

Extra suggestions for the manuscript;

In the introduction section;

- The following line; 40, 44 and 45 seem to imply that the author is studying people who wish to die at home and people’s experiences using health care services. In the rest of the manuscript, this is not the case. The author may wish to revise the above lines to reflect the specific problem that you are interrogating; ie the meaning of a good death.

- Line 69 seem to be incomplete

In the methodology section, please consider the following;

- The word both in line 85 seem to be inappropriate since the data sources are 3 not 2.

- The word intentional could be replaced with purposive in reference to the sampling procedure, if that is what the author wanted to mean. Check like 104 , 147 etc

- The author mentions that they excluded people did not have ‘powerful’ internet connection. The reader would be interested in knowing how they identified these- 164

- Throughout the manuscript, the author talks about discussion groups and not Focus Group Discussions. Is that intentional?

In the results section, please consider the following;

- In line 221, the author clearly shows that participants considered a good death in relation to a good life. One cannot have a good death unless they had a good life. It would be important for the author to explain what a good life was from the perspective of the study participants

In the conclusion, the author introduces a new concept of ‘red lines’ in relation to what participants refer to as unbearable suffering. However, it is important that the author introduces these concepts earlier in the manuscript to avoid new information in the conclusion section of the manuscript.

6. PLOS authors have the option to publish the peer review history of their article (what does this mean?). If published, this will include your full peer review and any attached files.

Do you want your identity to be public for this peer review? For information about this choice, including consent withdrawal, please see our Privacy Policy.

Reviewer #1: No

Reviewer #2: No

Reviewers' comments:

Reviewer's Responses to Questions

**Comments to the Author**

1. Is the manuscript technically sound, and do the data support the conclusions?

Reviewer #1: Partly

Reviewer #2: Partly

2. Has the statistical analysis been performed appropriately and rigorously? 

Reviewer #1: I Don't Know

Reviewer #2: N/A

3. Have the authors made all data underlying the findings in their manuscript fully available?

Reviewer #1: No

Reviewer #2: Yes

4. Is the manuscript presented in an intelligible fashion and written in standard English?

Reviewer #1: Yes

Reviewer #2: No

5. Review Comments to the Author

Reviewer #1: At this point, I have not yet had time to prepare the level of comments that I feel would be adequate to share with the authors. I hope that the discussion provided in the comments for the editor are enough to help you to decide either to give me an extension or to make a decision without my further input.

Reviewer #2: The manuscript would benefit from language editing. In some cases, the language needs to be checked to suit the standard language used in research.

Extra suggestions for the manuscript;

In the introduction section;

- The following line; 40, 44 and 45 seem to imply that the author is studying people who wish to die at home and people’s experiences using health care services. In the rest of the manuscript, this is not the case. The author may wish to revise the above lines to reflect the specific problem that you are interrogating; ie the meaning of a good death.

- Line 69 seem to be incomplete

In the methodology section, please consider the following;

- The word both in line 85 seem to be inappropriate since the data sources are 3 not 2.

- The word intentional could be replaced with purposive in reference to the sampling procedure, if that is what the author wanted to mean. Check like 104 , 147 etc

- The author mentions that they excluded people did not have ‘powerful’ internet connection. The reader would be interested in knowing how they identified these- 164

- Throughout the manuscript, the author talks about discussion groups and not Focus Group Discussions. Is that intentional?

In the results section, please consider the following;

- In line 221, the author clearly shows that participants considered a good death in relation to a good life. One cannot have a good death unless they had a good life. It would be important for the author to explain what a good life was from the perspective of the study participants

In the conclusion, the author introduces a new concept of ‘red lines’ in relation to what participants refer to as unbearable suffering. However, it is important that the author introduces these concepts earlier in the manuscript to avoid new information in the conclusion section of the manuscript.

6. PLOS authors have the option to publish the peer review history of their article (what does this mean?). If published, this will include your full peer review and any attached files.

Reviewer #1: **Yes: **Peter tamas

Reviewer #2: No

---

## [Author Response · Author response to Decision Letter 0]

30 Oct 2023

PLOS ONE

Note: HTML markup is below. Please do not edit.]

Journal Requirements:

Answer to editorial team: The authors have followed the style requirements of PLOS ONE.

"The study is part of a governmental commissioned project of the General Directorate of Health of the Catalan Health Department from the Government of Catalonia (Spain) in the context of the implementation of the Death Observatory to the Agency for Quality in Healthcare and Assessment of Catalonia (AQuAS)."

a) Please clarify the sources of funding (financial or material support) for your study. List the grants or organizations that supported your study, including funding received from our institution.

Answer to editorial team: As mentioned above, the study is part of a commissioned project but did not receive any specific funding or grant for its implementation. 

Answer to editorial team: The commissioners had no role in study design, data collection and analysis, the decision to publish or preparation of the manuscrit. Nevertheless, the research protocol and preliminary findings were discussed with the commissioners from the Department of Health of Catalonia to increase usefulness and impact for knowledge transfer and decision-making. 

Answer to editorial team: None of the coauthors received a salary from the commissioners. Lluïsa Fernández Giner received a salary from the Agency for HealthCare Quality and Assessment of Catalonia (AQuAS) as external and independent consultant paid through a general butged of internal comissioned project linked to the Death Observatory of Catalonia (Good death study code: 0002.LE01.20.037). 

Answer to editorial team: The authors received no specific funding for this work. 

Answer to editorial team: The narratives of participants and verbatims/quotes supporting the main insights of the study are freely available on the website of AQuAS, the public governmental institution dedicated to evaluating the quality of healthcare and development of decision-making tools/knowledge transfer.

For further information, please consult: 

https://aquas.gencat.cat/web/.content/minisite/aquas/publicacions/2022/estudi_qualitatiu_bona_mort_annexos_aquas2022.pdf

Answer to editorial team: Additional materials will be made available on acceptance of the manuscript in PLOS ONE. 

Answer to editorial team: Captions have now been added in appendix material.

Reviewers' comments:

Reviewer's Responses to Questions

Comments to the Author

1. Is the manuscript technically sound, and do the data support the conclusions?

Reviewer #1: Partly

Reviewer #2: Partly

Answer to reviewers: We have improved the manuscript by including additional verbatims/quotations that support the main findings and conclusions. 

2. Has the statistical analysis been performed appropriately and rigorously?

Reviewer #1: I Don't Know

Reviewer #2: N/A

Answer to reviewers: The project applied a qualitative ethnographical and phenomenological approach; as no quantitative data were obtained, no statistical analysis was performed.

3. Have the authors made all data underlying the findings in their manuscript fully available?

Reviewer #1: No

Reviewer #2: Yes

Answer to reviewers: The narratives of participants and main verbatims/ quotes supporting the main insights of the study are freely available on the website of AQuAS, the public governmental institution dedicated to evaluating the quality of healthcare and development of decision-making tools/ knowledge transfer.

For further information, please consult: 

https://aquas.gencat.cat/web/.content/minisite/aquas/publicacions/2022/estudi_qualitatiu_bona_mort_annexos_aquas2022.pdf

4. Is the manuscript presented in an intelligible fashion and written in standard English?

Reviewer #1: Yes

Reviewer #2: Yes

Answer to reviewers: A preliminary version of the manuscript (before first submission to PLOS ONE) was revised by a native professional biomedical translator. After including the changes suggested by the editorial team and reviewers, the same translator has revised the new version. 

5. Review Comments to the Author

Reviewer #1:

When looking at the text, I don’t see any obvious red flags. I, however, don’t see the information in the text that I require to be able to make an adequately detailed decision. I was hoping that I would find, attached somewhere, an annex that would allow me to understand exactly what they had done and why. 

Answer to reviewer 1: We thank the reviewer for this suggestion and have added a new appendix: 

APPENDIX 1. Techniques and conceptual-methodological approaches used in the qualitative study of a ”good death” in Catalonia (Spain). 

In response to the suggestions of reviewers, the previous Appendix 1 has been replaced.

I do believe that at the end of my review process I will come to the conclusion that what I’m seeing appears to be sound, but that the initial structure of the article does not adequately anticipate the methodology section, so it requires substantial revision. What I mean by this is that the choices made in the methodology section require information which should be, but is not present in the introduction. 

Answer to reviewer 1: In response to this comment we now discuss the rationale of the methodological approach in more detail in the introduction (paragraphs 1 to 3 of the new version on pages 4 and paragraph 1 on page 6).

I also believe that I will come to the conclusion that the method which they used is not a discourse analysis. As I have often found, I suspect that what I will find, when I have the opportunity to review more detailed information, is that they did a sound thematic content analysis. 

Answer to reviewer 1: We carried both a thematic content analysis and a discourse analysis. The interviews and focus group discussion were based on semistructured interviews. After each interview and focus group discussion a pre-analysis was carried out of the transcriptions, and selected verbatims were classified according to the main thematic topics described in table 1 of the manuscript. The main insights that emerged from each interview and groups were discussed by two researchers and further indepth interpretative discourse analysis allowed us to consider the opinions and experiences of patients, relatives and professionals. In the case of professionals, the narratives were considered according to their profiles. To aid in the description of the findings a summary of the main insights is presented. More information regarding the qualitative analysis carried out can be found in the technical report, available in the public repository of our governmental institution.

The study in general included the following main topics: 

-Presentation and life story of the interviewee

-Experience and impact of being sick/ getting old (either oneself or a family member) 

-Perceptions and meaning of a good death 

-Projecting what a good death is 

-Resources needed for achieving a good death 

We presented participants with images from nature. The participant chose one that they thought would represent a good death. They were also asked about the barriers and facilators for achieving a good death and about the unmet needs in the health system. In the case of professionals, we asked about the aspects they thought could be modified to improve the end of life and facilitate a good death. The perceptions of patients/people needing chronic or palliative care, the elderly, close family members or healthcare professionals were analysed both separately and globally. 

It is a constant frustration of mine that authors in my field, feel, properly, compelled to claim that they are doing analysis methods which are far more complex than that either possible or necessary for their purposes.

Answer to reviewer 1: Three theoretical-methodological perspectives were included in the present qualitative study to broaden our understanding of the meaning of a good death, of barriers and facilitators for achieving it, and also unmet needs. The present paper focused on the the concept of a good death; other findings of the broader study can be consulted elsewhere. 

The combination of an ethnographical perspective requiring a more naturalístic and less interpretative analysis with a more hermeneutical approach was carried out in the case of in depth interviews with patients and families. In the case of focus group discussions with professionals, a phenomenological approach was applied. The dynamics of the focus groups included a presentation of the main insights provided by patients and relatives in the study and debate/discussion. The analysis included a discourse analysis with a combination of different degrees of interpretation. The consideration of previously published studies and the discussion of findings with an independent advisory committee aided the validation of the main insights and the verbatims, and gave support to the findings and conclusions made. 

The main goal was to understand what a good death is, taking into consideration multiple participant profiles. This approach facilitates triangulation and enriches the analysis and validation of the findings.

I will find that the article failed to follow the expectations of annotation for transparent inquiry, which means that I do not understand the logic which they have used in sampling quotations for presentation in their text. 

Answer to reviewer 1: We agree with the reviewer, and now include additional quotations in the appendix of the new version of the manuscript. We have deleted the old appendix 1, and added new quotes that corroborate more clearly the main findings presented in the paper. We have included more details on the differences according to participant profiles in appendix 2A and 3.

Due to limitations of space and the specific goals defined in this manuscript, we chose to select the main common elements of a projected good death in global terms. 

I will also find that they failed to remain consistent in their recognition that the object, good death, would be understood very differently by their respondents. What this means, in consequence, is that rather than present, a set of scenarios at the end, each some sort of idealized representation selected for centrality. 

Answer to reviewer 1: We agree to some extent with the reviewer. As mentioned in the first paragraphs, the answer to the question “is the concept of a good death universal?” is “no”; it is a subjective concept, and the insights gained from the discourses of participants suggest that it should be defined as “dying in the way each individual person wants”. As in other published studies, and to aid the understanding of an abstract, subjective concept, we identified certain common elements, which were then revised and discussed further by the research team and the external independent steering group. Barriers and facilitators and unmet needs were also recorded and described elsewhere, and help to understand what the public health system can do to improve end-of-life care and planning. 

The new appendix 3 includes further findings regarding core elements of a good death for patients, relatives and professionals, as well as a table comparing core elements in the present study with those of previous studies that have applied qualitative or quantiative methodological approaches (appendix 4). 

What they have done is present a list of common

---

## [Decision Letter · Decision Letter 1]

12 Nov 2023

PONE-D-23-14275R1What is a good death for patients, families and health professionals? A qualitative ethnographic and phenomenological study in the Catalan contextPLOS ONE

Dear Dr. Serra-Sutton,

Thank you for submitting your manuscript to PLOS ONE. After careful consideration, we feel that it has merit but does not fully meet PLOS ONE’s publication criteria as it currently stands. Therefore, we invite you to submit a revised version of the manuscript that addresses the points raised during the review process.

We look forward to receiving your revised manuscript.

Kind regards,

Martin Mbonye

Academic Editor

PLOS ONE

Additional Editor Comments:

Dear Author,

I am glad to inform you that we have new comments to the revised version of your manuscript. Thank you very much for the efforts you put into this manuscript, but there is still some more major revisions for you to address before we can consider this manuscript ready for publication. I would advise that you look carefully at the comments and provide a detailed response.

It appeared to the reviewer that your response to some comments was not detailed enough and did not provide enough to justify some of the claims you made particularly on methodology. This is where most of the issues seem to stem from.

I do advise that as the author(s), you carefully read through the reviewer's comments and respond to each of them in good detail.

Please address the comments by 7th December, 2023. If you want more time, please let me know before that deadline

Regards and all the best

Martin

Reviewers' comments:

Reviewer's Responses to Questions

**Comments to the Author**

1. If the authors have adequately addressed your comments raised in a previous round of review and you feel that this manuscript is now acceptable for publication, you may indicate that here to bypass the “Comments to the Author” section, enter your conflict of interest statement in the “Confidential to Editor” section, and submit your "Accept" recommendation.

Reviewer #1: (No Response)

2. Is the manuscript technically sound, and do the data support the conclusions?

Reviewer #1: No

3. Has the statistical analysis been performed appropriately and rigorously? 

Reviewer #1: No

4. Have the authors made all data underlying the findings in their manuscript fully available?

Reviewer #1: No

5. Is the manuscript presented in an intelligible fashion and written in standard English?

Reviewer #1: Yes

6. Review Comments to the Author

Reviewer #1: 1. Is the manuscript technically sound, and do the data support the conclusions?

Reviewer #1: Partly

Reviewer #2: Partly

2. Has the statistical analysis been performed appropriately and rigorously?

Reviewer #1: I Don't Know

Reviewer #2: N/A

3. Have the authors made all data underlying the findings in their manuscript fully available?

Reviewer #1: No

Reviewer #2: Yes

Comment on revision: the PDF does not meet my understanding of the requirements for either archiving or data. The data are not in an archive, the relationship between the data found and the recording is uncertain, there are no meta-data and there is no qualitative equivalent to the R script I would expect to find with archived data

5. Review Comments to the Author

Reviewer #1:

When looking at the text, I don’t see any obvious red flags. I, however, don’t see the information in the text that I require to be able to make an adequately detailed decision. I was hoping that I would find, attached somewhere, an annex that would allow me to understand exactly what they had done and why.

• Comment on revision: The level of documentation of the methodology and methods remains inadequate.

I do believe that at the end of my review process I will come to the conclusion that what I’m seeing appears to be sound, but that the initial structure of the article does not adequately anticipate the methodology section, so it requires substantial revision.

• Comment on revision: the changes made have degraded my assessment of the rigor of analysis.

What I mean by this is that the choices made in the methodology section require information which should be, but is not present in the introduction.

• Comment on revision: Much of the information required to understand the logic behind selection of methods and the details of their use remain missing.

I also believe that I will come to the conclusion that the method which they used is not a discourse analysis.

• Comment on revision: I still lack the information required to determine whether the analysis conducted qualifies as a discourse analysis. For that I require a defensible definition followed by analytic practice that clearly matches that definition.

As I have often found, I suspect that what I will find, when I have the opportunity to review more detailed information, is that they did a sound thematic contact analysis. It is a constant frustration of mine that authors in my field, feel, properly, compelled to claim that they are doing analysis methods which are far more complex than that either possible or necessary for their purposes.

• Comment on revision: the information provided did not improve insight into the steps by which analysis was undertaken. For example, the terms ‘vertical and then horizontal reading’ do not mean much without both a definition and justification. This journal is read by a wide diversity of readers. They can not be expected to know what these are nor whether they are appropriate. This information must be provided by the authors.

I will find that the article failed to follow the expectations of annotation for transparent inquiry, which means that I do not understand the logic which they have used in sampling quotations for presentation in their text.

• Comment on revision: the authors did not improve the transparency through mechanisms such as ATI.

I will also find that they failed to remain consistent in their recognition that the object, good death, would be understood very differently by their respondents. What this means, in consequence, is that rather than present, a set of scenarios at the end, each some sort of idealized representation selected for centrality. What they have done is present a list of common elements across respondents that are chosen more for commonality than salience. This sort of presentation of an intersection of subjective constructions is a common strategy that is used which suffers the unfortunate characteristic of failing to remember that each of the construction is considered may fail horribly when shorn of attributes specific to that single construction.

• Comment on revision: The authors seem to flip-flop between accepting the possibility of a heterogeneity of mutually incommensurate constructions of ‘the good death’ and the policy convenience of a determinate list of identically understood (and therefore, somehow, important) themes.

7. PLOS authors have the option to publish the peer review history of their article (what does this mean?). If published, this will include your full peer review and any attached files.

Reviewer #1: **Yes: **Peter A. Tamas

---

## [Author Response · Author response to Decision Letter 1]

3 May 2024

Reviewers' comments:

Reviewer's Responses to Questions

Comments to the Author

1. If the authors have adequately addressed your comments raised in a previous round of review and you feel that this manuscript is now acceptable for publication, you may indicate that here to bypass the “Comments to the Author” section, enter your conflict of interest statement in the “Confidential to Editor” section, and submit your "Accept" recommendation.

Reviewer #1: (No Response). 

AUTHOR’S ANSWERS: The authors appreciate the feedback received and considers have made the manuscript gain clarity and soundness. After reviewing the suggestions to improve, we hope to have answered your requirements for publication of the manuscript base on a qualitative phenomenological study design.

2. Is the manuscript technically sound, and do the data support the conclusions?

Reviewer #1: No

AUTHOR’S ANSWERS: We have added more details in the manuscript section and appendix 1 to increase soundness.

3. Has the statistical analysis been performed appropriately and rigorously? 

Reviewer #1: No

AUTHOR’S ANSWERS: the basis of this study is qualitative, including opinions, perceptions and experiences. In this case, no quantitative data were collected, so no statistical analysis were expected or needed. 

4. Have the authors made all data underlying the findings in their manuscript fully available?

Reviewer #1: No

AUTHOR’S ANSWERS: the full transcripts of interviews and discussion groups are available in Catalan and can be requested to the research team. A complete narrative can be consulted online in the website of the Agency for Health Quality and Assessment of Catalonia (AQuAS) and Sciencia Salut repository. 

5. Is the manuscript presented in an intelligible fashion and written in standard English?

Reviewer #1: Yes

AUTHOR’S ANSWERS: we appreciate this consideration. A revision of a native professional English translator has been made to a previous version.

6. Review Comments to the Author

Reviewer #1: 1. Is the manuscript technically sound, and do the data support the conclusions?

Reviewer #1: Partly

Reviewer #2: Partly

AUTHOR’S ANSWERS: In a previous version sent before this second round of reviews, we had added additional information in appendixes document. The reference to this key material has been added in the main manuscript more explicitly, offering more soundness and transparent description of the process made to support main data insights and conclusions made. 

It should be considered that several Ethical committees approved the research protocol and the interviews followed predefined guidelines as mentioned in methods section. All interviews and focus group discussions were transcripted and analysed to identify different and common elements of a good death. Main insights were discussed among the main researchers during all the fieldwork and in different stages of analysis process. Furthermore, the findings are similar among different profiles of participants, techniques applied and the published literature at International level gaining consistency and validity. 

We have revised and improved the new version of the manuscript, including more information in the methodology, references to additional quotings to support insights and published studies with similar findings to our study. This study has contributed to gain father understanding of what is an ideal good death to aid decision-makers and citizens in general as to express needs at end of life. 

2. Has the statistical analysis been performed appropriately and rigorously?

Reviewer #1: I Don't Know

Reviewer #2: N/A

AUTHOR’S ANSWERS: the basis of this study is qualitative, based on opinions, perceptions and experiences. In this case, no quantitative data were collected, so no statistical analysis were expected or needed. 

3. Have the authors made all data underlying the findings in their manuscript fully available?

Reviewer #1: No

Reviewer #2: Yes

Comment on revision: the PDF does not meet my understanding of the requirements for either archiving or data. The data are not in an archive, the relationship between the data found and the recording is uncertain, there are no meta-data and there is no qualitative equivalent to the R script I would expect to find with archived data

AUTHOR’S ANSWERS: Each interview was trascripted. These materials are available in Catalan. A report was produced (grey literature) was produced and includes an appendix document available in our website with more detail of narratives and guidelines for the interview conduction: 

https://observatorisalut.gencat.cat/ca/observatori_mort/estudi-de-la-bona-mort/

https://aquas.gencat.cat/web/.content/minisite/aquas/publicacions/2022/estudi_qualitatiu_bona_mort_annexos_aquas2022.pdf

https://scientiasalut.gencat.cat/handle/11351/8275

Further information of narratives and data can be requested to the main corresponding author. 

5. Review Comments to the Author

Reviewer #1:

When looking at the text, I don’t see any obvious red flags. I, however, don’t see the information in the text that I require to be able to make an adequately detailed decision. I was hoping that I would find, attached somewhere, an annex that would allow me to understand exactly what they had done and why.

• Comment on revision: The level of documentation of the methodology and methods remains inadequate.

I do believe that at the end of my review process I will come to the conclusion that what I’m seeing appears to be sound, but that the initial structure of the article does not adequately anticipate the methodology section, so it requires substantial revision.

AUTHOR’S ANSWERS: We appreciate this comment. Due to space limitation, we wrote a summary of methods followed. As mentioned above, the research protocol was approved by the Ethics Committee SIDIAP Jordi Gol (approval code in methods section) and this research protocol has also followed quality criteria of research of our Agency for Healthcare Quality and Research of Catalonia (AQuAS). We have included an additional Appendix 1 to describe what we did, when and how. 

• Comment on revision: the changes made have degraded my assessment of the rigor of analysis.

What I mean by this is that the choices made in the methodology section require information which should be, but is not present in the introduction.

AUTHOR’S ANSWERS: we have added further rationale of the analysis carried out in the methods section of the new manuscript being submitted. 

• Comment on revision: Much of the information required to understand the logic behind selection of methods and the details of their use remain missing.

I also believe that I will come to the conclusion that the method which they used is not a discourse analysis.

• Comment on revision: I still lack the information required to determine whether the analysis conducted qualifies as a discourse analysis. For that I require a defensible definition followed by analytic practice that clearly matches that definition.

As I have often found, I suspect that what I will find, when I have the opportunity to review more detailed information, is that they did a sound thematic contact analysis. It is a constant frustration of mine that authors in my field, feel, properly, compelled to claim that they are doing analysis methods which are far more complex than that either possible or necessary for their purposes.

AUTHOR’S ANSWERS: we carried out both a content analysis and discourse analysis of patient’s, close family relative and professionals involved in care and clinical management view’s, opinions and experiences. The basis of this study is a phenomenological qualitative approach aiming to understand the study topic, good death. It was considered fundamental the voices and perpective of people with chronic health problems, older people without any chronic complex or advanced chronic condition but vitally nearer the end of their life, caregivers and also professionals perspectives. So, both descriptive content analysis and more interpretative discourse analysis in order to understand discourses according to different narratives and profiles was considered. Due to the sensitive topic being treated in the study, images of nature were used (please find in appendix of our technical grey literature report) as icebreaker and aid participants in expressing what their ideal death (good death was). In their case in depth, qualitative interviews in the house of participants were made. The interviewer applied an agreed guideline and interviews were recorded to facilitate transcripts and pre-analysis for validity reasons. The approach was considered hermeneutical also as there was the need to understand the phenomenon and experiences in the natural way interviews felt the concept/phenomenon being studied. We agree with the reviewer that even if the natural context of interviews was included it cannot be strictly considered ethnographical. The analysis included a certain degree of interpretation taking into account an inductive approach, considering the discourses of informants and views into account as a way to understand in more depth the meaning of the concept, barriers, facilitators and needs for different populations (ex. those in the end-of life phase, family members or professionals that attend them). The positioning of the interviewer was not participant as in ethnographic approach and no participant observations were made.

• Comment on revision: the information provided did not improve insight into the steps by which analysis was undertaken. For example, the terms ‘vertical and then horizontal reading’ do not mean much without both a definition and justification. This journal is read by a wide diversity of readers. They can not be expected to know what these are nor whether they are appropriate. This information must be provided by the authors.

I will find that the article failed to follow the expectations of annotation for transparent inquiry, which means that I do not understand the logic which they have used in sampling quotations for presentation in their text.

AUTHOR’S ANSWERS: we have added more clarity in the methods section on the analysis carried out and more detail of approach in appendix 1. Additionally, more quotings have been included in Appendix 2, to complement insights in the main manuscript to increase robustness of findings. 

• Comment on revision: the authors did not improve the transparency through mechanisms such as ATI.

I will also find that they failed to remain consistent in their recognition that the object, good death, would be understood very differently by their respondents. What this means, in consequence, is that rather than present, a set of scenarios at the end, each some sort of idealized representation selected for centrality. What they have done is present a list of common elements across respondents that are chosen more for commonality than salience. This sort of presentation of an intersection of subjective constructions is a common strategy that is used which suffers the unfortunate characteristic of failing to remember that each of the construction is considered may fail horribly when shorn of attributes specific to that single construction.

• Comment on revision: The authors seem to flip-flop between accepting the possibility of a heterogeneity of mutually incommensurate constructions of ‘the good death’ and the policy convenience of a determinate list of identically understood (and therefore, somehow, important) themes.

AUTHOR’S ANSWERS: we appreciate this consideration. As in other studies mentioned in the Introduction, Discussion and now Appendix 4, the authors of this study have found the subjectivity round the positioning of what is a good death, expressed in the first section of results and discussion. Nevertheless, the goal was to understand and be concreate on what would be an ideal good death, first as a way of expressing preferences and needs for patients, family and citizens in general, gain specificity of the barriers and facilitators (these findings not presented here due to space reasons and publication elsewhere) and specially what the system (health, and other sectors) can do in specific terms to: break taboos round death, meet the preferences and expressed needs. The next steps would be to create care and community based standards to meet what is considered and ideal of good death. The results again has similarities and differences among participats, expressed in the present paper, and again with published studies. Unfortunately, and due to strong bioethical considerations, several ethical committees were requested and most participants were people who were willing to talk openly about good death. This is a limitation, mentioned in discussion section as people not willing to participate still can confront taboo in front of this phenomenon, good death, death and end of life planning. To be as independent as possible, different scientific societies from health sector and patient associations, together with key representatives of different care providers participated from the start in this commissioned project. 

7. PLOS authors have the option to publish the peer review history of their article (what does this mean?). If published, this will include your full peer review and any attached files.

Do you want your identity to be public for this peer review? For informa

---

## [Editor Report · Decision Letter 2]

2 Jun 2024

PONE-D-23-14275R2What is a good death for patients, families and health professionals? A qualitative phenomenological study in the Catalan contextPLOS ONE

Dear Dr. Serra-Sutton,

Thank you for submitting your manuscript to PLOS ONE. After careful consideration, we feel that it has merit but does not fully meet PLOS ONE’s publication criteria as it currently stands. Therefore, we invite you to submit a revised version of the manuscript that addresses the points raised during the review process. Please see below under the section additional Editor's comments, the summary of the issues that you need to address

We look forward to receiving your revised manuscript.

Kind regards,

Martin Mbonye

Academic Editor

PLOS ONE

Journal Requirements:

Additional Editor Comments:

Dear Author,

Thank you very much for your patience with this paper. It is almost there but there are a few things that were suggested by the reviewer that I feel still need your attention as well as a few things. Below are the areas that you should look at and address before we can proceed.

1. Under materials and Methods: The reviewer was not satisfied that you had addressed in your revision of the methods section. The reviewer suggested that you give more details and justify the selection of the approach you use. I suggest that instead of pointing the reader to another location for details that instead you carefully read the comments on this particular issue and add more details.

2. Under sampling: Please be clear on which sample was selected under each of your sampling procedures. For example, who was purposively selected and who was selected by convenience sampling and why?

3. Under materials: Say something on the method used of Phenomenology in order to satisfy the reviewer's misgivings about clarity of methods. Also show why it was the most suited for answering your research question for readers unfamiliar with the concept.

4. I suggest that you include a section of the setting so that the non-Spanish readers appreciate the (rural and ethnic) context. You mention Catalonia but it may not be universally known

5. Under data analysis: Please respond to the reviewer's comment on finding a definition of discourse analysis and how you applied the discourse analysis based on your definition

6. The reviewer was not satisfied that your use of the terms vertical and horizontal under the analysis section added much value. So I suggest that you explain them clearly or replace them with more orthodox terms

7. You were asked to explain transparancy in application of participants quotes. I think a way around this would be to say something in the methods section indicating your use of quotes. Read any document you can find on annotation for transparency inquiry (ATI) for guidance

8. Lastly, there are still a number of gramatical errors and typos in the revised version. Please use an English speaker to help with the editor after you are done
---

## [Author Response · Author response to Decision Letter 2]

3 Oct 2024

Please find answers to editorial team and reviewers in the attached document "Answers to editors and reviewers",

---

## [Editor Report · Decision Letter 3]

8 Oct 2024

Defining a Good Death: Perspectives of Patients, relatives, and Health Care Professionals in the Catalan Context—a Qualitative Study

PONE-D-23-14275R3

Dear Dr. Serra-Sutton,

We’re pleased to inform you that your manuscript has been judged scientifically suitable for publication and will be formally accepted for publication once it meets all outstanding technical requirements.

Kind regards,

Martin Mbonye

Academic Editor

PLOS ONE

Additional Editor Comments (optional):

I would advise you to use the remaining period to carefully read through and correct any typos, areas with double full stops, areas that need grammatical improvements. These should make the paper more readable and error free
---

## [Editor Report · Acceptance letter]

18 Oct 2024

PONE-D-23-14275R3 

PLOS ONE

Dear Dr. Serra-Sutton, 

I'm pleased to inform you that your manuscript has been deemed suitable for publication in PLOS ONE. Congratulations! Your manuscript is now being handed over to our production team.

Kind regards, 

on behalf of

Dr. Martin Mbonye 

Academic Editor

PLOS ONE